# Efficient Bayesian inference for mechanistic modelling with high-throughput data

**Simon Martina Perez**[1] *, **Heba Sailem**[2]℗, **Ruth E. Baker**[1]℗

**1** Mathematical Institute, University of Oxford, Oxford, United Kingdom, **2** Institute of Biomedical Engineering Science, University of Oxford, Oxford, United Kingdom

℗ These authors contributed equally to this work.
* martinaperez@maths.ox.ac.uk

**Data Availability Statement:** All code to generate synthetic data, as well as code used to analyse the data is available on Github at https://github.com/simonmape/minibatch-ABC. All data needed to

## Abstract

Bayesian methods are routinely used to combine experimental data with detailed mathematical models to obtain insights into physical phenomena. However, the computational cost of Bayesian computation with detailed models has been a notorious problem. Moreover, while high-throughput data presents opportunities to calibrate sophisticated models, comparing large amounts of data with model simulations quickly becomes computationally prohibitive. Inspired by the method of Stochastic Gradient Descent, we propose a *minibatch* approach to approximate Bayesian computation. Through a case study of a high-throughput imaging scratch assay experiment, we show that reliable inference can be performed at a fraction of the computational cost of a traditional Bayesian inference scheme. By applying a detailed mathematical model of single cell motility, proliferation and death to a data set of 118 gene knockdowns, we characterise functional subgroups of gene knockdowns, each displaying its own typical combination of local cell density-dependent and -independent motility and proliferation patterns. By comparing these patterns to experimental measurements of cell counts and wound closure, we find that density-dependent interactions play a crucial role in the process of wound healing.

## Author summary

During wound healing, cells work together to close a wound to restore tissue integrity. Thousands of different genes play a role in wound healing, and scratch assay experiments are routinely used to investigate the role of these genes by analysing how a wound closes when each of these is not expressed, *i.e.* knocked down. So far, the impact of knocking down genes on wound healing has been determined by comparing the size of the wound before and after a given time period, but these measurements do not elucidate the fine-scale mechanisms that determine how cells behave in the presence of their neighbours. By combining a detailed mathematical model with experimental imaging of wound healing, we identify how cells respond to and work together with their neighbours during wound healing. Applying this method to a large number of gene knockdowns, we identify three well-defined functional subgroups of knockdowns, each displaying its own typical behaviours of movement and proliferation to close the wound. These observations explain the

reproduce the study can be found on our Zenodo repository: DOI 10.5281/zenodo.5898532.

**Funding:** S.M.P. is supported by an EPSRC/UKRI Doctoral Training Award (https://www.ukri.org). H. S. is supported by Sir Henry Wellcome Fellowship grant 204724/Z/16/Z (https://wellcome.org). R.E.B. would like to thank the Royal Society for a Wolfson Research Merit Award (https://royalsociety.org). The funders had no role in study design, data collection and analysis, decision to publish, or preparation of the manuscript.

**Competing interests:** The authors have declared that no competing interests exist.

role of each of the knockdowns on wound healing and further our understanding of cell-cell interactions in wound healing.

# 1 Introduction

High-throughput methods entail the acquisition and processing of vast amounts of experimental data with ever increasing detail. This wealth of data creates unique opportunities to derive insights about real-world phenomena. However, the quantitative metrics used for analysis of high-throughput data sets often do not exploit the full spatial and temporal information in the data. For example, in scratch assay experiments routinely used in cell biology studies, typically only cell count and wound area are used to interpret outcomes [1, 2]. Such straight-forward summary statistics can identify biologically relevant differences between different experimental conditions, but they are not generally suited to identify complex spatial and temporal phenomena that arise from *e.g.* interactions between individual cells. Further, while the ever increasing availability of detailed experimental data has the potential to inform, calibrate and refine complicated mathematical models that incorporate such cell-cell interactions, the computational cost of this process quickly becomes a bottleneck when the amount of data to be analysed is increased. In this work, we seek to develop a method to combine mathematical modelling and high-throughput data to extract detailed, real-world insights from a large data set with a fraction of the computational cost of existing methods.

Mathematical modelling and experimental data are routinely combined using Bayesian inference, and of late there has been intense interest in both the opportunities offered by high-throughput data and models (see for instance the editorial overview by Hasenauer and Banga [3]) as well as the required protocols for combining mathematical modeling and high-throughput data successfully [4]. Given a mathematical model and observed data, Bayesian methods express the information gained from data in probability distributions for the parameters that constitute the model. Such *posterior distributions* provide information as to which model parameters reproduce observed data well while also expressing the associated uncertainty in these values given the data. One of the challenges in performing Bayesian analysis of complex mathematical models is that the likelihood function of simulated data given a parameter value is typically intractable, so that simulation-based approaches to likelihood estimation are required. This means that large amounts of computationally costly simulations are required to compare model outputs with experimental observations, which is complicated further when the parameter set of the model is high-dimensional, as the cost of accurate likelihood estimation scales poorly with the dimension of the parameter space [5, 6]. The computational cost of likelihood-free Bayesian inference methods has prompted a huge volume of research into improving the computational efficiency of Bayesian methods [7], such as multifidelity methods [8], optimal perturbation kernels [9, 10], and delayed acceptance [11] to name but a few.

The central challenge for likelihood-free Bayesian inference with high-throughput data is that in practice one might need to repeatedly simulate a new instance of the model for each of the observed data points in order to accurately estimate the likelihood. This may be necessary, for instance, when the data consist of time series with highly variable initial conditions: a reasonable comparison between simulated and observed data can then only be made when each observed and simulated data point share the same initial condition. When the number of data points is large, such as with high-throughput data, the resulting computational cost becomes prohibitive, as the simulation cost grows linearly with the amount of data, in addition to growing exponentially with the dimension of the parameter space. Even when other state-of-the-art

approaches are used, the fundamental bottleneck of repeated simulation remains. Inspired by the method of stochastic gradient descent (SGD) [12–14] in machine learning, we propose a *minibatch* approach to tackle this issue: for each comparison between simulated and observed data, we use a stochastically sampled subset (minibatch) of the data. A similar minibatch method has been employed very recently by Stapor *et al.* [15] to successfully calibrate ordinary differential equation (ODE) models with a significant improvement in computational performance, and by Seita *et al.* [16] within the context of MCMC, likewise with a significant computational speed-up. We demonstrate that choosing a large enough minibatch ensures that the relevant signatures in the observed data can be accurately estimated, while avoiding unnecessary comparisons that slow down inference. We apply our minibatch approach to perform Approximate Bayesian computation (ABC) with a well-established stochastic individual-based model (IBM) of density-dependent cell migration, proliferation, and death [17, 18] on the high-throughput data set from Williams *et al.*'s scratch assay screen [19]. The pipeline is illustrated in Fig 1 scratch assays involve growing a cell monolayer to confluence and then mechanically removing (scratching away) a portion of the cell population to leave an "in vitro wound". Typically, a small region around the scratched area is imaged to provide information on the dynamics of wound closure [20]. Such assays are a simple, fast, and inexpensive method to investigate collective cell invasion under varying environmental or genetic conditions.

In the screen of Williams *et al.* [19], human dermal lymphatic endothelial cells (HDLECs) were used, both in a control setting and after knockdown of some 500 genes using RNA interference.Williams *et al.* [19] identified the effect of different gene knockdowns on wound healing by comparing the extent of wound closure after a period of 24 hours. While these metrics

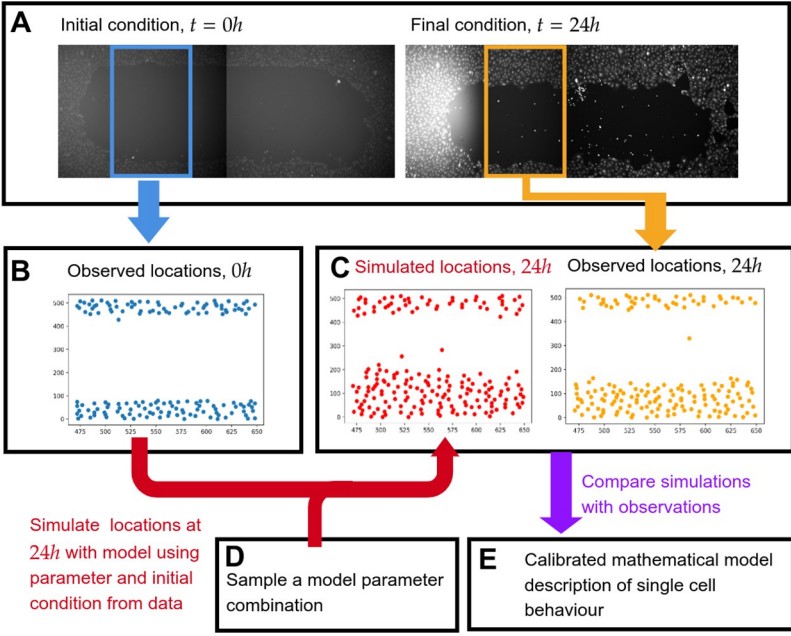

**Fig 1. Schematic representation of the process of combining mechanistic modelling and scratch assay data.** A: each observation in the data set consists of an initial condition at $t = 0h$ and a final condition at $t = 24h$. B: the field of view of the initial condition is cropped and cell locations are extracted. C: the field of view of the final condition is cropped and cell locations are extracted, as for the initial condition. A mechanistic mathematical model is used to simulate cell locations at $t = 24h$ using a sampled parameter from box D. E: by simulating many times with different parameter values we can compare which parameters generate simulated data that is mathematically similar to observations.

are informative of general, large-scale, features of collective cell invasion, they do not intrinsically reflect the mechanisms that allow the different knockdowns to display different wound healing behaviours. In this study, we use computational Bayesian inference with an interpretable, mechanistic model to obtain detailed descriptions of cell-cell interactions. By calculating the posterior parameter distribution of the model for each gene knockdown, we characterise the functional behaviour of each perturbation according to the cell-cell interaction and cell-intrinsic mechanisms of the model. By analysing the extent of similarity between different gene knockdowns, we identify three major functional subgroups, which differ mainly according to their intrinsic and density-dependent movement rates. To our knowledge, this is the first time a mechanistic model has been used to interpret and classify high-throughput data to elucidate functional subgroups.

The structure of this work is as follows. In Section 2, we describe the high-throughput data, the IBM and the proposed minibatch Bayesian inference scheme. In Section 3, we characterise the performance of the method using different batch sizes and apply it to characterise functional mechanistic differences between different genetic perturbations. We conclude in Section 4 with a discussion of our results. All code and data to reproduce the study are publicly available at https://github.com/simonmape/minibatch-ABC and our Zenodo repository: DOI 10.5281/zenodo.5898532.

## 2 Methods and model

In this section, we detail the experimental data that will be used in this work (Section 2.1), outline the computational model (Section 2.2), and the Bayesian method used to combine model and data (Section 2.3).

### 2.1 Data from a gene knockdown screen

We used a previously published high-throughput siRNA screen of wound healing in HDLECs [19] where more than 500 gene knockdowns were identified as significant for altering wound healing [19]. During the wound healing experiment, images of the wound were captured at $t = 0h$, $24h$ after the scratch was made. Images of cells were acquired at 4x objective, which allowed the entire experimental well to be captured in two images that were stitched together, resulting in $512 \times 1392$-pixel images, where each pixel corresponds to $2.97\mu m$ [19].

We used DeepScratch [21] to analyse image data from various gene knockdowns. DeepScratch is a neural network with U-NET architecture. It detects cell centres from raw cell images stained with markers of the nucleus or membrane, and in addition it uses the locations of detected cells to segment the wound as the area that is void of cells. Following image analysis using DeepScratch, we computed various features including the distance of each of the cells to the wound edge and cell density. As the image data lack a cytoplasmic stain, we approximated cell area by generating a Voronoi tessellation where a polygon is defined for each cell centre. Cell locations are then given in a rectangular grid with dimensions $512 \times 1392$ pixels. Only assays where both an initial ($t = 0h$) and final condition ($t = 24h$) are present in the data set are included, and we also only include assays where both wound edges are contained in the initial condition. For each image, a wound mask was computed to isolate the extent of the wound and exclude dead and extruded cells. We describe the filtering process in detail in Supplementary Information S1 Text Section S1. To exclude noise from measurements at the location where images are stitched together, and to obtain data where we only consider movement perpendicular to the wound edge, we further reduce the field of view to a $512 \times 180$-pixel subset of the image, 10 pixels left of where the images where stitched together. Applying both filters to the data set yields 118 candidate gene knockdowns for investigation.

## 2.2 An individual-based mechanistic model of density-dependent cell movement, proliferation, and death

In this work, we use a well-established stochastic, lattice-free IBM developed by Binny *et al.* [18] to model density-dependent cell migration, proliferation, and death. This model has previously been shown by Browning *et al.* [17] to accurately represent collective cell migration in a wound healing assay experiment. Density-dependent behaviour in the model is incorporated through cell-to-cell interactions, so that behaviour depends on local crowding [17]. Following [18], we let $\boldsymbol{x}_n = (x_n, y_n)$ be the location of cell $n$, where $n = 1, \ldots, N(t)$ and $N(t)$ is the population size at time $t$. Interactions between cells are governed by a measure of crowding, where each cell senses the cells in its close proximity. Mathematically, the crowding function at location $\boldsymbol{x}$ is given by

$$B(\boldsymbol{x}) = \sum_{i=1}^{N(t)} \gamma_b w(\|\boldsymbol{x} - \boldsymbol{x}_n\|^2), \tag{1}$$

where the function $w$ is an exponential kernel that quantifies the impact of local crowding upon cell behaviours. This specific choice of $w(r) = \exp(-r^2/2\sigma^2)$ has been found to give rise to outputs that match experimental data well, but other choices for $w$ can also be made [18]. To simplify computations, we assume that $w(r) = 0$ whenever $r \geq 3\sigma$. The parameter $\gamma_b > 0$ in Eq (1) describes the magnitude of the crowding effect, and $\sigma$ governs the length scale over which agents can interact with other agents. Browning *et al.* [17] suggest that a suitable choice for the interaction parameter $\sigma$ is $\sigma = \varphi/2$, where $\varphi$ is the cell diameter. In this work, we will estimate the cell diameter, $\varphi$, from experimental data by using the measured cell areas in the images acquired at $t = 0h$ and $t = 24h$. In Supplementary Information S1 Text Section S5, we report this estimation procedure in detail.

The crowding function, $B$, governs the direction in which agents prefer to move and proliferate: agents prefer to move and proliferate along the steepest descent of $B$. The direction of steepest descent is also called the *bias vector*. For the $n$-th agent, we define the bias vector as

$$\boldsymbol{B}_n = -\nabla B(\boldsymbol{x}_n). \tag{2}$$

When a cell attempts to move in the model, the direction of movement, $\phi$, is sampled from the von Mises distribution as

$$\phi \sim \text{von Mises}(\arg(\boldsymbol{B}_n), \|\boldsymbol{B}_n\|). \tag{3}$$

The rationale for this is that $\mathbb{E}[\phi] = \arg(\boldsymbol{B}_n)$, meaning that the direction of movement and proliferation is on average in the direction of the bias vector, and $\phi \sim \mathcal{U}(0, 2\pi)$ in the limit as $\|\boldsymbol{B}_n\| \to 0$, meaning that cells in isolation exhibit unbiased movement.

Cell movement, proliferation, and death occur according to a Poisson process, which we simulate using the Gillespie algorithm [22]. Each cell's movement and proliferation rates depend on the extent of local crowding, whereas we let the death rate be constant and independent of crowding. In the Supplementary Information S1 Text we explore the effect of including a density-dependent death term in the model. We compute Bayes factors for both models using the resulting posteriors [23–25] to find that the data contains stronger evidence in favour of density-independent death. The movement rate for the $n$-th cell is denoted by $M_n$, and the proliferation rate for the $n$-th cell is denoted by $P_n$. Each individual rate consists of a base, intrinsic rate, which is then modulated by the cell's interactions with its neighbours. The

movement and proliferation rates for cell $n$ are given by

$$M_n \;\;= \max\left\{0, m - \gamma_m \sum_{i=1, i \neq n}^{N(t)} \exp\left(-\frac{\|\boldsymbol{x}_n - \boldsymbol{x}_i\|^2}{2\sigma^2}\right)\right\}, \tag{4}$$

$$P_n \;\;= \max\left\{0, p - \gamma_p \sum_{i=1, i \neq n}^{N(t)} \exp\left(-\frac{\|\boldsymbol{x}_n - \boldsymbol{x}_i\|^2}{2\sigma^2}\right)\right\}, \tag{5}$$

and the death rate is a constant $d > 0$. The parameters $\gamma_m$ and $\gamma_p$ modulate the strength of a cell's response to crowding. For example, when $\gamma_m < 0$, local crowding increases motility, when $\gamma_m = 0$ motility is independent of local density, and when $\gamma_m > 0$ motility decreases with local crowding.

When a movement event occurs for any given cell, that cell moves one cell diameter, $\varphi$, in the direction prescribed by the distribution of the angle $\phi$. When a proliferation event occurs instead, the cell places a daughter cell one cell diameter away in the direction of $\phi$. We assume periodic boundary conditions to model the fact that the field of view is a small subset of a much larger tissue. In our implementation, we choose the domain to match the field of view of the cropped high-throughput imaging data. Hence, we simulate on a domain of size $180 \times 512$ pixels.

## 2.3 Bayesian statistics: Connecting models and data

In this work, we are interested in how observed collective behaviours within the scratch assays might be described in terms of the mechanistic properties of individual cells. That is, given observed data of wound closure in the scratch assay, we wish to understand which parameters $\boldsymbol{\theta} = (m, p, d, \gamma_m, \gamma_p, \gamma_b)$ might make such observations probable. The goal of Bayesian parameter estimation is to update prior beliefs about the model parameters of the IBM from Section 2.2 with real-world data. Prior beliefs are encoded in a prior distribution $\pi(\boldsymbol{\theta})$, which defines a probability density function for parameters $\boldsymbol{\theta}$. Given a parameter vector $\boldsymbol{\theta}$, the model defines a density function for the observations, which, when normalised, is called the likelihood, $P(\mathcal{D}_{\mathrm{obs}} \,|\, \boldsymbol{\theta})$. Bayes' rule allows one to combine the likelihood with the prior distribution to give the posterior distribution:

$$P(\boldsymbol{\theta} \,|\, \mathcal{D}_{\mathrm{obs}}) \propto P(\mathcal{D}_{\mathrm{obs}} \,|\, \boldsymbol{\theta})\pi(\boldsymbol{\theta}). \tag{6}$$

By using Bayes' rule, the prior is updated with data, which in this work consists of the repeated measurements of cell centre locations at $t = 0h$ and $t = 24h$. In this work, we choose independent uniform priors for each of the parameters, and we let the prior ranges be informed by the findings of Browning *et al.* [17] so that $m \sim \mathcal{U}(0, 10)h^{-1}$, $p, d \sim \mathcal{U}(0, 0.05)h^{-1}$, $\gamma_b \sim \mathcal{U}(-2.5, 2.5)h^{-1}$, $\gamma_p \sim \mathcal{U}(0, 0.05)h^{-1}$, and $\gamma_b \sim \mathcal{U}(0, 50)\mu m$. We note that we estimate the cell area directly using DeepScratch, rather than assigning a prior distribution and performing Bayesian inference, as has been the standard approach in previous works, such as Browning *et al.* [17]. To understand the reliability of our results, in the Supplementary Information S1 Text we investigated the sensitivity of the posterior distributions to the input cell area parameter and find that for values of the cell area within three standard deviations of the median cell estimate, relative differences between the resulting posterior means are within 10% of the parameter value found using the estimated cell size, suggesting that the model predictions are robust. At the same time, we observe that a large variation exists in the measured cell areas in the high-throughput screen, such that it becomes unwise to assign a single value for cell area to all the

knockdowns in the screen. Further, we note that assigning a single value for the cell area parameter leads to unreliable predictions.

**2.3.1 Approximate Bayesian computation.** In practice, the likelihood for complex stochastic models is mathematically intractable and typically only accessible through simulation. Approximate Bayesian computation is a popular likelihood-free tool to estimate the left-hand side in Eq (6) [24, 26]. It approximates the likelihood, $P(\mathcal{D}_{\mathrm{obs}} \mid \boldsymbol{\theta})$, using repeated simulation of the model, accepting simulated parameters $\boldsymbol{\theta}$ only if the resulting model output, $\mathcal{D}_{\mathrm{sim}}(\boldsymbol{\theta})$, is closer than some threshold, $\epsilon$, to the data, $\mathcal{D}_{\mathrm{obs}}$:

$$P_{\mathrm{ABC}}(\boldsymbol{\theta} \mid \mathcal{D}_{\mathrm{obs}}) \propto P(d(\mathcal{D}_{\mathrm{obs}}, \mathcal{D}_{\mathrm{sim}}) < \boldsymbol{\varepsilon} \mid \boldsymbol{\theta})\pi(\boldsymbol{\theta}), \qquad (7)$$

where $d$ is some distance function that quantifies the difference between the data, $\mathcal{D}_{\mathrm{obs}}$, and model output, $\mathcal{D}_{\mathrm{sim}}$. In many applications of ABC, the data are high-dimensional, meaning that, from a computational point of view, it is necessary to summarise the data in lower-dimensional summary statistics so that the distance $d(\mathcal{D}_{\mathrm{obs}}, \mathcal{D}_{\mathrm{sim}})$ in Eq (7) can be efficiently computed. In Section 2.3.3, we describe our choice of summary statistic for this problem.

We use a version of sequential Monte Carlo sampling (ABC-SMC) [24, 26] as our implementation of ABC. In brief, ABC-SMC propagates a sample from the prior distribution, $\pi(\boldsymbol{\theta})$, through a sequence of intermediate distributions to approximate the target distribution, $P(\boldsymbol{\theta}|\mathcal{D}_{\mathrm{obs}})$. The intermediate distributions are a sequence of ABC approximations $P_t$ for $t \in \{1, \ldots, T\}$. We use a sequential importance sampling approach to ABC-SMC [24, 27, 28], where each Monte Carlo sample $\{\boldsymbol{\theta}_i^{(t)}, w_i^{(t)}\}$ built at generation $t$ is used to construct an importance distribution at generation $t + 1$. We define $N_{\mathrm{gen}}$ as the total number of parameters that are drawn from the importance distribution at each generation, and $N_{\mathrm{acc}}$ as the total number of parameters that are accepted. Given a Monte Carlo sample at generation $t$, the sample at generation $t + 1$ is obtained by using a *perturbation kernel $K(\cdot|t)$* to propose $N_{\mathrm{gen}}$ perturbed parameters from the Monte Carlo sample at generation $t$, from which the $N_{\mathrm{acc}}$ proposed parameters with smallest distance to the real-world data are accepted into the Monte Carlo sample for generation $t + 1$. In this work, we use the Normal perturbation kernel proposed by Filippi *et al.* [9] and let $N_{\mathrm{gen}} = 2 \cdot 10^4$ and $N_{\mathrm{acc}} = 500$, meaning that each generation has an acceptance rate of 2.5%. The final sample, $\{\boldsymbol{\theta}_i^{(T)}, w_i^{(T)}\}$, forms the output of the ABC-SMC algorithm. All implementation details are outlined in Supplementary Information S1 Text Section S2.

**2.3.2 Minibatch-ABC-SMC.** In some problems, one model simulation can be compared with arbitrarily many observations, using the distance function $d$. This is particularly the case when the model is deterministic and one realisation of the model suffices to capture the relevant behaviour of the model. When the model is stochastic, however, a single model realisation will likely not provide sufficient information on the distribution of model outputs to sensibly compare to observed data. An additional, and significant, problem occurs with time series data because the initial conditions across different experiments can be vastly different. This means that for each comparison with a different datapoint, a new model simulation with a specific initial condition is needed to ensure that the comparison between the summary statistics of the simulated and observed data is fair and informative. For high-throughput data, the implications of these additional computational costs are significant. As an illustration of this point, the data set in this work consists of many hundreds of observations with different initial wound location, initial cell densities and initial topology. Fig 2 illustrates the extent to which two wells aimed at studying the same genetic profile can differ. To apply Bayesian inference approaches to combine models to such data requires individual simulations, which mirror the initial conditions of each experiment separately.

**Fig 2. Two initial conditions from the assays conducted on wild-type cells.** Note the variation in both the location of the wound as well as the size of the initial wound, highlighting the need to carry out separate simulations, one for each initial condition.

When observed data, $\mathcal{D}_{\mathrm{obs}} = \{y^i_{\mathrm{obs}}\}^{N_{\mathrm{obs}}}_{i=1}$, can only be compared to model simulations that share the same initial condition, the simulated data will equally consist of $N_{\mathrm{obs}}$ datapoints, such that $\mathcal{D}_{\mathrm{sim}} = \{y^i_{\mathrm{sim}}\}^{N_{\mathrm{obs}}}_{i=1}$, and $y^i_{\mathrm{sim}}$ shares its initial condition with $y^i_{\mathrm{obs}}$. A scalar measure of the discrepancy between the observed and simulated data is then obtained by taking the mean of distances between the observed data points and their corresponding model simulations:

$$d(\mathcal{D}_{\mathrm{sim}}, \mathcal{D}_{\mathrm{obs}}) = \frac{1}{N_{\mathrm{obs}}} \sum_{i=1}^{N_{\mathrm{obs}}} d(y^i_{\mathrm{sim}}, y^i_{\mathrm{obs}}). \tag{8}$$

When the size of observed data, $N_{\mathrm{obs}}$, is large, *i.e.* $N_{\mathrm{obs}} \gg 1$, and the model needs to be simulated separately for each data point, ABC-SMC approaches fail to scale well with the number of observations: simulation time—which is normally a bottleneck—increases linearly with the number of observations.

Drawing inspiration from minibatch methods in SGD in machine learning [12–14], we propose a minibatch approach to computing the distance in Eq (8): rather than using all $N_{\mathrm{obs}}$ observations to compute the distance whenever a parameter $\boldsymbol{\theta}$ is proposed in ABC-SMC, we suggest computing the distance between observed and simulated data based on randomly chosen *batches* from the data. This is done by choosing a *batch size*, $N_{\mathrm{bs}}$, such that $N_{\mathrm{bs}} \ll N_{\mathrm{obs}}$, that controls the number of datapoints considered in each comparison between model simulations and data. Formally, this amounts to drawing $N_{\mathrm{bs}}$ random numbers $i_j$ independently with replacement, from $\{1, \ldots, N_{\mathrm{obs}}\}$, obtaining the corresponding data point $y^{i_j}_{\mathrm{obs}}$ and simulating a model output, $y^{i_j}_{\mathrm{sim}}$, that shares the same initial condition. Then, the discrepancy between model and data can be approximated by

$$d(\mathcal{D}_{\mathrm{sim}}, \mathcal{D}_{\mathrm{obs}}) \approx \frac{1}{N_{\mathrm{bs}}} \sum_{j=1}^{N_{\mathrm{bs}}} d(y^{i_j}_{\mathrm{sim}}, y^{i_j}_{\mathrm{obs}}). \tag{9}$$

This approach immediately reduces simulation time by a factor $N_{\mathrm{obs}}/N_{\mathrm{bs}}$. The minibatch approach is simple to implement within the importance sampling part of the ABC-SMC framework described in Section 2.3.1. At each generation, $t$, an importance distribution is given (for $t = 1$ this is the prior distribution). Then, $N_{\mathrm{gen}}$ parameters are sampled according to this distribution. For each parameter a minibatch of size $N_{\mathrm{bs}}$ is sampled from the data and for each observation within that minibatch a corresponding model simulation is generated given the parameter and observed initial condition. Mathematically, if $y^{\xi}_{\mathrm{obs}}(0)$ is the initial condition corresponding to observed data point $y^{\xi}_{\mathrm{obs}}$, the model simulates a corresponding output $y_{\mathrm{sim}} \sim f(\bullet|\boldsymbol{\theta}_n, y^{\xi}_{\mathrm{obs}}(0))$. The discrepancy between simulated and observed data is computed for each parameter using Eq (9). The algorithm is summarized in Algorithm 1.

**Algorithm 1**: Minibatch-importance sampling

**Input:** Data, $\mathcal{D}_{\text{obs}} = \{y_{\text{obs}}^i\}_{i=1}^{N_{\text{obs}}}$; number of parameters to sample, $N_{\text{gen}}$; number of parameters to accept, $N_{\text{acc}}$; batch size, $N_{\text{bs}}$; prior distribution, $\pi(\boldsymbol{\theta})$; importance distribution, $\hat{q}(\boldsymbol{\theta})$; model, $f(\bullet \mid \boldsymbol{\theta})$; distance function, $d$.

**Output:** $N_{\text{acc}}$ weighted samples $\{\boldsymbol{\theta}_n, w_n\}_{n=1}^{N_{\text{acc}}}$.

```
1  for i = 0, ..., N_gen do
2    Increment i ← i + 1;
3    Generate parameter vector θ_i according to given importance distri-
       bution q̂(θ);
4    for k = 1, ..., N_bs do
5      Draw data index ξ uniformly at random from {1, ..., N_obs};
6      Simulate y_sim ∼ f(• | θ_n, y_obs^ξ(0));
7    end
8    Calculate ε_i using Eq (9).
9  end
10 From {θ_i, w_i}_{n=1}^{N_gen}, select the N_acc parameters with lowest value of ε_i;
11 Set w_i = π(θ_i)/q(θ_i) for all accepted parameters θ_i.
```

At this point, we remark that an alternative to using minibatch ABC-SMC might be to *a priori* select a subset of the data (*e.g.*, using stratified sampling) and use ABC-SMC on this subset. While reducing the amount of data to be processed would certainly reduce the computational burden of ABC-SMC, this approach also vastly reduces the amount of available information. Moreover, subsampling *a priori* from the data might introduce bias into the problem, as it is up to the practitioner to define a sensible subset of the data. In Supplementary Information S1 Text Section S4.5, we carefully examine the quality of the posteriors generated by subsampling from the data in comparison to those generated using minibatch ABC-SMC. We find that *a priori* selected subsets give rise to more variability in the estimated posterior mean, and higher posterior variance across all parameters, minibatch ABC-SMC with minibatches of the same size. We conclude that our approach is to be preferred over simply reducing the amount of available information by subsampling the data.

Our method contains the minibatch size, $N_{\text{bs}}$, and the number of generations, $T$, as tunable hyperparameters. In machine learning applications, it has been noted that inference with smaller batch sizes when optimizing an objective function leads to less overfitting to the data, as the algorithm is better able to observe the full range of parameter values that yield results that resemble the full distribution of the data, rather than the parameters that only produce summary statistics that are close to the average observed in the data set [13, 14]. At the same time, using a very small batch size might not provide a good estimate of the model distribution, meaning that careful consideration of the batch size is necessary. This will be the focus of Section 3.1. For the number of generations, we choose $T = 4$. This hyperparameter choice comes from performing test runs of minibatch ABC-SMC with various batch sizes and numbers of generations and computing the Kullback-Leibler (KL) divergence [29] between the successive posteriors as a measure of change of the distribution from one generation to another, as is done by Filippi *et al.* [9]. As a general stopping criterion, we suggest to choose the smallest $T$ so that the KL divergences for the subsequent generations converge to a fixed value. In this specific problem, given the model and the data, this amounts to $T = 4$, indicating that the distributions do not change significantly after this generation. In other problems, a different number of generations will typically be needed. For details on how the KL divergence between successive generations was calculated, we refer the reader to Supplementary Information S1 Text Section S2.3.

We note that the random sampling of minibatches ensures that with very high probability every datapoint in the dataset will be used in each generation. Since for each of the $N_{\text{gen}}$

parameters in each generation there are $N_{bs}$ datapoints sampled uniformly at random, the probability that one generation of minibatch ABC-SMC fails to contain a given datapoint, say $i_\star \in \{1, \ldots, N_{obs}\}$, can be computed. Denoting this event $E_{i_\star}$, a simple computation shows that this probability can be bounded by

$$\mathbb{P}(E_{i_\star}) = \left(1 - \frac{1}{N_{obs}}\right)^{N_{gen} \cdot N_{bs}} \leq e^{-(N_{gen} \cdot N_{bs})/N_{obs}}, \tag{10}$$

since $(1 + x/n)^n < e^x$ for any integer $n$. For $N_{gen} = 2 \cdot 10^4$,

$$\mathbb{P}(E_{i_\star}) \leq \exp\left(-\frac{2 \cdot N_{bs}}{N_{obs}} \cdot 10^4\right), \tag{11}$$

meaning that the event that in one generation not all datapoints are used is essentially negligble.

All simulations were carried out using a development branch of the parallel open source ABC-SMC software Pakman [30]. Each ABC-SMC experiment was performed on a high performance compute (HPC) cluster using five 48-core Cascade Lake (Intel Xeon Platinum 8268 CPU @ 2.90GHz) nodes.

**2.3.3 Summary statistics.**    To compute the distance $d$ between observed and simulated data, we employ the summary statistics proposed by Browning *et al.* [17], with the aim to capture three key features: population size increase, spatial structure and density profile [17]. To capture population size increase, we use the population size, $N(t)$, at $t = 24h$. To capture spatial structure, we use the pair correlation function, $P(r, t)$, which describes the density of pairs of agents separated by a distance $r$ at time $t$, relative to the expected density of pairs if the population were uniformly distributed [17]. For our discrete data, the pair correlation function describes the relative number of pairs separated by distances in the range $(j - 1)\Delta r < r < j\Delta r$. Following [17], we compute

$$P(j, t) = \frac{LH}{N(t)^2 \pi \Delta r(2j + \Delta r)} \sum_{n=1}^{N(t)} \sum_{i=1, i\neq n}^{N(t)} \mathbb{I}((j - 1)\Delta r < \|x_i - x_n\| < j\Delta r). \tag{12}$$

In Eq (12), $L$ and $H$ are length and height, respectively, of the field of view. Following the recommendation in [17], we choose $\Delta r = 5\mu m$, and calculate the pair correlation function up to a distance of $100\mu m$. To capture the density profile, we compute the density across crosssections of the wound by dividing the field of view into 64 different bins in the direction parallel to the scratch and computing the cell density in each bin. Using 64 bins results in a bin width of approximately $24\mu m$, or, one cell diameter. Assigning a centroid for each cell, and using this centroid to define the location of cell $i$ as $y_i$, this results in a one-dimensional density $\rho(j, t)$ at time $t$, where

$$\rho(j, t) = \sum_{i=0}^{N(t)} \mathbb{I}(j\Delta y \leq y_i < (j + 1)\Delta j). \tag{13}$$

These three distance functions allow us to define the distance between an observed data point $y_{obs}^i$ and a simulated data point $y_{sim}^i$ as

$$d(y_{sim}^i, y_{obs}^i) = \frac{[N_{sim}(24) - N_{obs}(24)]^2}{N_{obs}(24)^2} \quad + \frac{\sum_{j=1}^{20} [P_{sim}(j, 24) - P_{obs}(j, 24)]^2}{\sum_{j=1}^{20} P_{obs}(j, 24)^2}$$
$$+ \frac{\sum_{j=1}^{64} [\rho_{sim}(j, 24) - \rho_{obs}(j, 24)]^2}{\sum_{j=1}^{64} \rho_{obs}(j, 24)^2}. \tag{14}$$

## 3 Results

In this section, we demonstrate that minibatch ABC-SMC is accurate and efficient at obtaining posterior distributions over model parameters given the high-throughput data, and that it can be used to shed light on the mechanistic impact of a wide range of genetic perturbations. In Subsection 3.1, we explore how the choice of hyperparameters in the algorithm, as well as the quantity of available information, affects the reliability of the inference process so that we can confidently apply the minibatch ABC-SMC algorithm to data from the high-throughput screen. In Subsection 3.2, we show how the IBM can be used in tandem with minibatch ACB-SMC to detect functional differences between different gene knockdowns. In Subsection 3.3, we use minibatch ABC-SMC on the data from 118 gene knockdowns to locate each genetic perturbation in the five dimensional parameter space of the IBM, according to its estimated posterior mean. Analysing the spatial structure of the locations of each gene knockdown in this space allows us to distinguish a range of functionally different subgroups within the data that represent genetic perturbations with functionally similar behaviours. Given the biological interpretation of the different model parameters, this allows us to reach conclusions about the primary drivers of wound closure in the different genetic perturbations. Importantly, our analysis shows the importance of density-dependent effects on observed changes in cell count number during the experiment.

### 3.1 Small batch sizes provide reliable inference

The introduction of minibatches in ABC-SMC comes with the task of selecting the batch size as an algorithm hyperparameter. A good choice for the batch size, $N_{\text{bs}}$, needs to balance computational complexity on the one hand and the generation of a reliable estimate of the discrepancy between model and data on the other hand. To understand the capacity of the minibatch ABC-SMC algorithm to distinguish different model parameter values from data in the context of the IBM, we generate two synthetic datasets, each corresponding to a different parameter regime. Parameter regime I corresponds to a high movement rate with a positive impact of density on movement and a low proliferation rate, such that $m = 1.5h^{-1}$, $p - d = 0.01h^{-1}$, $\gamma_m = -1h^{-1}$, $\gamma_p = 0.01h^{-1}$ and $\gamma_b = 20\mu m$. Parameter regime II corresponds to a low movement rate with contact inhibition for movement and a high proliferation rate, such that $m = 0.5h^{-1}$, $p - d = 0.025h^{-1}$, $\gamma_m = 1.5h^{-1}$, $\gamma_p = 0.01h^{-1}$ and $\gamma_b = 20\mu m$. For each parameter regime, an *in silico* data set is created by simulating one model output at $t = 24h$ for every initial condition corresponding to Mock. That is, each of the two *in silico* datasets contains 117 data points, each corresponding to one initial datum from the high-throughput experiment. Then, we perform minibatch-ABC-SMC on the synthetic datasets, varying the batch size, such that $N_{\text{bs}} = 1, 5, 10, 50, 117$. For each batch size, we repeat the inference process five times.

 **3.1.1 Variability of the posterior distributions.** In Supplementary Information S1 Text Section S4, we report the posterior means of the resulting posterior distributions, and their variance, at each batch size, as well as representative plots of the posterior distributions. At any value of the batch size, minibatch ABC-SMC correctly identifies the features of each parameter regime. For instance, for the dataset for parameter regime I, the estimated intrinsic motility rate, $m$, is consistently estimated as higher than that of the dataset for parameter regime II. Conversely, the net intrinsic proliferation rate, $p - d$, is correctly estimated as higher in parameter regime II than in parameter regime I. For the density-dependent interaction term, $\gamma_m$, the minibatch ABC-SMC implementation discerned at any batch size that the effect of crowding on motility is positive in regime I, and negative in regime II. Increasing the batch size improved the quality of the posterior distributions, although the improvements diminish

rapidly as the batch size, $N_{bs}$, is increased. There is a critical value—in Fig E in S1 Text this is typically around $N_{bs} = 10$—after which increasing the batch size, $N_{bs}$, does not futher reduce the variability of the estimated posterior means. Similarly, after this critical value, the capacity of minibatch ABC-SMC to identify the true posterior mean does not improve, even when the batch size is increased to the full data size. Similarly, the standard deviation of the posterior distributions diminishes rapidly when the batch size is increased initially, but then stays relatively constant across larger batch sizes. In both parameter regimes, this is the case for $N_{bs} = 10$. Taken together, analysis of the posterior means and posterior standard deviations suggests that a minibatch implementation of ABC-SMC provides a robust approximation of the posterior distribution at a fraction of the computational cost. In the synthetic data set, the computational speedup of using a batch size of 10 samples compared to the full data set amounts to a factor $117/10 = 11.7$.

**3.1.2 Reliability of minibatch ABC-SMC using experimental data.** Finally, to compare performance on real-life data from the scratch assay, we perform inference on the Mock type data set in S1 Text Section S4, where we vary the batch size from 1 to 117 and find no meaningful differences between the different batch sizes in terms of their ability to identify the posterior means consistently. With growing batch size, just as with synthetic data, the variance of the posterior distribution decreases and shows a similar behaviour to the synthetic control data around a batch size of $N_{bs} = 10$. We conclude that choosing a minibatch size of $N_{bs} = 10$ allows for minibatch ABC-SMC to reliably identify the posterior mean.

While some genetic perturbations have a large number of observations and the choice of $N_{bs} = 10$ offers a good balance between computational complexity, robust inference of the posterior mean and low variance of the posterior distribution, some perturbations have very few observations in the high-throughput screen used in this work. As such we address the suitability of ABC-SMC in identifying mechanistically relevant differences between the different gene knockdowns by selecting 40 datapoints from each of the Mock and CDH5 knockdown experiments at random and without replacement, and dividing them into 20 pairs of two observations for each of the datasets. We perform ABC-SMC on each of the resulting *miniature data sets* and record their posterior distributions. We find that while the spread of the posteriors is indeed larger for each of the experimental conditions than in the case when more datasets are included, the posterior means can still identify meaningful differences between the two different genetic perturbations. This suggests that as few as just two observations can be used to confidently estimate model parameters.

Finally, we assess how well our IBM predicts experimentally observed wound healing behaviour when parametrised according to the posteriors generated using minibatch ABC-SMC. We sample 1000 parameters from the posterior distribution obtained using minibatch ABC-SMC with $N_{bs} = 10$ on the full Mock dataset and select one Mock initial condition uniformly at random. For this condition, we perform forward model simulations, one with each sampled parameter, and record the summary statistics defined in Section 2.3.3, which are the density profile, pair correlation function, and number of alive cells at $t = 24h$. From these forward simulations, we compute the mean and standard deviation for the density profile and pair correlation function. In Fig 3, we display the experimental data for density profile and pair correlation together with a range of one standard interval from the posterior mean. We also show a histogram of simulated cell counts at $t = 24h$, compared to the experimentally observed cell count at $t = 24h$. The posterior predictive ranges for the density profile and pair correlation are excellent fits to the data, and the simulated cell count histogram is centered around the experimentally observed cell count.

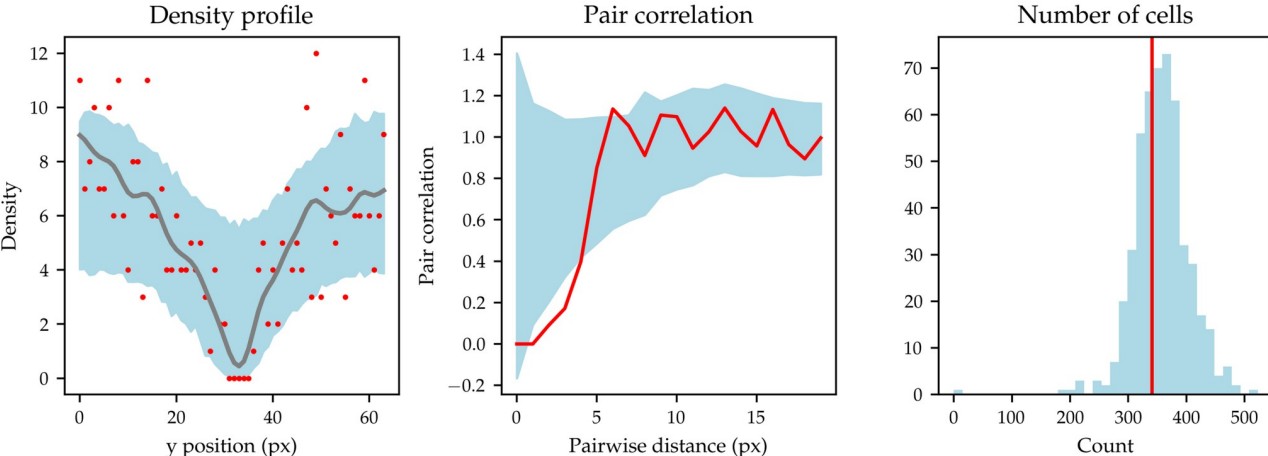

**Fig 3. Posterior predictive check for the model.** Left: experimentally observed density profile for a single datapoint sampled at random from the Mock dataset (red points) displayed against the posterior range of one standard deviation away from the posterior mean (blue shaded region). The gray line is the smoothed density profile generated using a Gaussian kernel. Middle: experimental pair correlation function (red line) displayed against the posterior range away of one standard deviation away from the posterior mean (blue shaded region). Right: experimentally observed cell count at $t = 24h$ (red vertical line) compared to the histogram of model simulations of cell count (blue). All summary statistics have an excellent fit to the experimental data.

## 3.2 Detecting differences between different experimental conditions

Our data set contains three genetic perturbations with a large number of observations. These are: Mock, CDC42 knockdown and CDH5 knockdown. For each of the observations, we apply the minibatch ABC-SMC algorithm using the implementation details from Section 2.3. We use the findings from Section 3.1 to choose a batch size $N_{bs} = 10$ for all experiments. We note that the total number of observations for Mock, CDC42 and CDH5 knockdowns are 117, 52 and 31, respectively, meaning that in the inference process for these experiments using $N_{bs} = 10$, minibatch ABC-SMC yields a speedup of roughly 12×, 5×, 3×, respectively.

Fig 4 reveals a striking difference between the three genetic profiles in terms of their intrinsic movement, intrinsic net proliferation and motility parameters. In Supplementary Information S1 Text Section S7, we calculate confidence intervals for the posterior means of each parameter and find that the confidence intervals are disjoint for intrinsic motility rate, $m$, net intrinsic proliferation rate, $p − d$, and density-dependent motility rate, $\gamma_m$, while only CDH5 has a confidence interval disjoint from CDC42 and Mock, respectively, in the density-dependent parameters $\gamma_p$ and $\gamma_b$. Of these genetic profiles, Mock knockdown is shown to have behaviour best described by strong contact-mediated motility, while motility is much less impacted by local cell density in the CDH5 knockdown case, and this effect is much smaller still in the CDC42 knockdown case. Further, Mock knockdown has the highest intrinsic net proliferation rate, whereas the CDH5 and CDC42 knockdowns are inferred to have lower net proliferation rates. In fact, the posterior distributions for the CDH5 and CDC42 knockdowns have support on negative values of $p − d$, which is consistent with the fact that some of the CDH5 and CDC42 knockdown assays show a decrease in the number of cells over the course of the experiment. In line with previous inference performed on this mechanistic model of density-dependent cell movement and proliferation, the parameters controlling density-dependent proliferation, $\gamma_p$, and the strength of density-dependent interactions, $\gamma_b$, are not conclusively identified from the data [17].

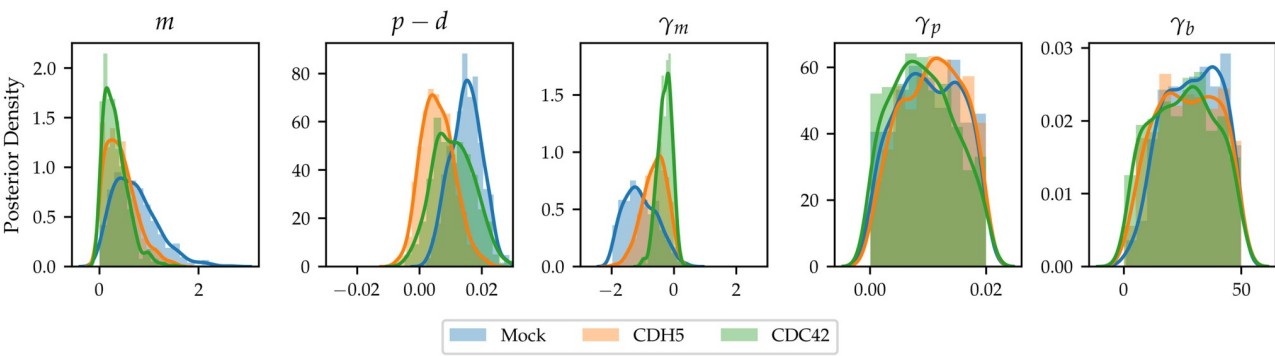

**Fig 4. Posterior distributions for three genetic perturbations with large numbers of observations.** Blue: Mock condition, orange: CDH5 knockdown, green: CDC42 knockdown.

Our findings are in agreement with previous experiments. For example, loss of CDC42 expression is associated with defective adhesion, wound healing, polarity establishment, and migration (see, for instance, the review by Mendelez *et al.* [31]), which is consistent with our finding that the CDC42 knockdown has less strong density-dependent migration than Mock: a loss of polarity establishment and adhesion is consistent with less directed movement into the wound and away from areas of high local tissue density. Likewise, CDH5 plays "a[n] important role in endothelial cell biology through control of the cohesion and organization of the intercellular junctions" [32], which is again consistent with a loss of local density-dependent mechanisms.

In addition to comparing *between* the different knockdowns in terms of their cellular migratory dynamics, the large amount of observations per knockdown in a high-throughput screen also enables inquiry into the factors that might create variation *within* any given genetic perturbation. For instance, the data for Mock and CDH5 knockdown contain very large variations in initial wound size, meaning that an important question that can be explored is the functional effect of initial wound size on migratory dynamics. It is known from the literature that changing the initial density of cells in a scratch assay has far-reaching consequences for the resulting dynamics [17]. In the same way, Jin *et al.* found that the initial geometry in experimental models of wound closing plays an important role in the wound closing dynamics [33], raising the possibility that the size of the wound could also have a strong effect on the wound-closure dynamics. To address this question, we measured the initial wound size for both the Mock and CDH5 knockdown datasets. (The CDC42 knockdown experiments had a much smaller variation in initial wound size, making it impossible to differentiate meaningfully between experiments based on initial wound size.) We find that initial wound sizes in the Mock dataset range between $\sim 1.79\text{mm}^2$ and $\sim 3.06\text{mm}^2$, and between $\sim 1.89\text{mm}^2$ and $\sim 2.7\text{mm}^2$ in the CDH5 knockdown dataset. We choose to divide the Mock data set into three different wound size categories: small ($< 2.43\text{mm}^2$), medium ($2.43\text{mm}^2 - 2.70\text{mm}^2$), and large ($> 2.70\text{mm}^2$). Similarly, we divide the CDH5 data set in two categories: small ($< 2.25\text{mm}^2$) and large ($> 2.25\text{mm}^2$). For each of the data subsets—large, medium, and small Mock; large and small CDH5 knockdown, respectively—we perform minibatch ABC-SMC with a batch size $N_{\text{bs}} = 10$ to identify the migratory dynamics within these conditions.

Fig 5 shows the marginal posterior distributions for model parameters for both the Mock and CDH5 knockdown datasets. To understand the large variance seen in Fig 5 for the intrinsic motility parameter, *m*, in the small wound category, we further analysed the wound closure data and observed that the smallest wounds at time $t = 0h$ in the Mock case were fully closed

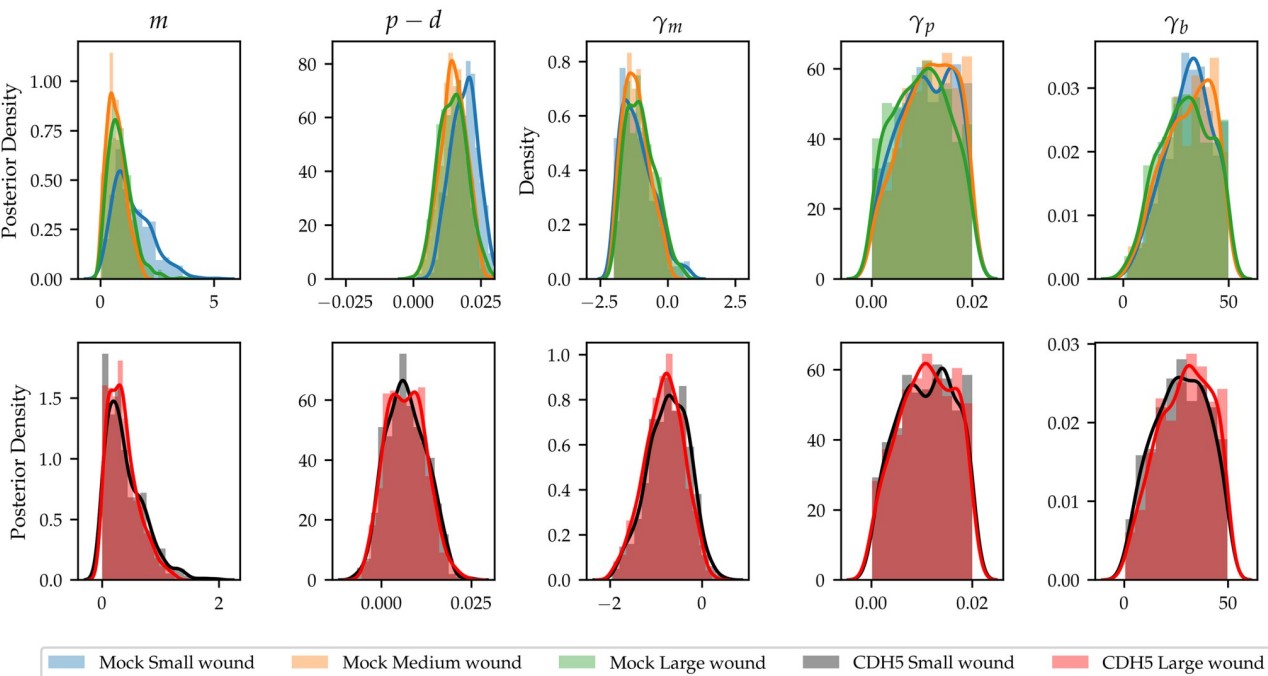

**Fig 5. Posterior distributions for model parameters when inference is carried out on subsets of the data based on wound size.** Top row: Mock experiments, three different wound size categories are identified—small (blue, $< 2.43\text{mm}^2$), middle (orange, $2.43\text{mm}^2 - 2.70\text{mm}^2$), and large (green, $> 2.70\text{mm}^2$). Bottom row: CDH5 knockdown, two different wound size categories are identified—small (black, $< 2.25\text{mm}^2$) and large (red, $> 2.25\text{mm}^2$).

by $t = 24h$. Further, when this is the case, the intrinsic motility parameter becomes practically unidentifiable: by simulating new model outputs using the IBM, we found that for any fixed value of the net intrinsic proliferation rate, $p - d$, and density-dependent movement parameter, $\gamma_m$, higher intrinsic movement rates did not influence the density or pair correlation summary statistics detailed in Section 2.3.3. Therefore, the higher variance of the posterior distribution in the small wound size category for the Mock data set is likely attributable to parameter non-identifiability given the available data, rather than to biologically significant mechanistic signatures in the data. We quantify the (dissimilarity) between the resulting marginal posterior distributions by computing confidence intervals for the posterior means in Supplementary Information S1 Text Section S7. We find that none of the confidence intervals, except for that of intrinsic motility rate, $m$, in Mock are disjoint from those of the other initial wound size categories. This implies that migratory dynamics for the different initial wound sizes are very similar.

## 3.3 High-throughput identification of mechanistic effects of gene knockdowns

Having established that minibatch ABC-SMC can accurately identify different mechanistic cell behaviours, we employ ABC-SMC on a large number of gene knockdowns in the high-throughput screen in order to assess and identify the range of different behaviours observed in the set of knockdown experiments. For each gene knockdown, a posterior distribution is obtained and each genetic perturbation is identified with a point in five-dimensional parameter space, given by the value of its posterior mean. We reiterate that there is a large variability in the number of available data points for each knockdown. Here, we consider all gene

knockdowns for which more than 2 data points exist in the data set, and for all gene knock-downs we choose $N_{\text{bs}} = 2$ accordingly.

**3.3.1 Clustering analysis reveals functional subgroups.** The set of posterior means for the different gene knockdowns is a set of points in five-dimensional parameter space, and it can be studied to reveal patterns of cell behaviours across the different gene knockdown experiments. One popular method of understanding the spatial structure of data, when it cannot be readily visualised due to the dimension of the data space, is *K*-means clustering. The idea behind *K*-means clustering is to divide a set of points in Euclidean space into *K clusters* such that the sum of the pairwise distances within each cluster is minimised. With other words, it is an unsupervised method to assign labels to the datapoints such that points with the same label are close to each other given some defined distance function. Mathematically, given *K* clusters, the classification problem is to subdivide the data into *K* distinct clusters, $\mathcal{S} = \{S_1, \ldots, S_K\}$, where the clusters are found by solving the optimization problem

$$\mathcal{S} = \arg \ \min \sum_{i=1}^{k} \frac{1}{2|S_i|} \sum_{\boldsymbol{x},\boldsymbol{y} \in S_i} \|\boldsymbol{x} - \boldsymbol{y}\|^2, \tag{15}$$

where $|S_i|$ denotes the number of points in cluster *i*, and $\|\bullet\|$ denotes the Euclidian distance. The number of clusters, *K*, is a hyperparameter that needs to be carefully chosen. Increasing the number of clusters will always decrease the total sum of pairwise distances, however it also leads to a loss of interpretability of the cluster labels and potentially overfits the data. A common practice in data science is to use the *elbow method* [34], which was first employed by Thorndike [35]. The elbow method amounts to choosing *K* as the elbow of the curve of the minimum of the objective function in Eq (15) as *K* is varied. The interpretation of picking the elbow of the curve, in clustering, corresponds to choosing *K* such that adding futher clusters does not provide a significantly better fit to the data. We perform *K*-means clustering for $K = 1, \ldots, 10$, and find that the marginal improvement of the sum of pairwise distances rapidly decreases after $K = 3$ (see Supplementary Information S1 Text Section S6). Therefore, we perform clustering with three clusters. To further analyse robustness, we assess the difference between clusters when we set $K = 2, 4, 5$ to find that the addition of extra clusters yields further sub-clustering within the clusters found at $K = 2$. From this we conclude that $K = 3$ is an good choice to balance mechanistic insight with more detailed spatial information.

The pairwise marginal distributions of the posterior means for the different gene knock-downs is shown in Fig 6, where the colour key is defined by the cluster labels. The separation between the clusters is best seen in the pairwise martinal posterior distribution of the intrinsic motility parameter, *m*, and the density-dependent motility parameter, $\gamma_m$. The marginal posterior distributions of the two parameters show that the cluster labels correspond to changes in the sign of the density-dependent motility parameter, $\gamma_m$, and the magnitude of the intrinsic motility parameter, *m*. Interpreting this separation mechanistically allows us to identify three functionally distinct clusters in the data. We identify Cluster I as knockdowns with low intrinsic motility, and the motility is enhanced by high local cell density (negative values for $\gamma_m$). In Cluster II, knockdowns have low intrinsic motility, and the motility is inhibited by high local cell density (positive values for $\gamma_m$). In Cluster III, knockdowns have high intrinsic motility, and the motility is enhanced by high local cell density (positive values for $\gamma_m$).

Furthermore, the pairwise plot of the intrinsic net proliferation parameter, $p - d$, and the density-dependent motility parameter, $\gamma_p$, shows that for knockdowns with a value of $p - d$ close to zero, there is a wide range of possible values for the density-dependent proliferation parameter, $\gamma_p$, whereas the variability of $\gamma_p$ diminishes for $p - d$ further away from zero. As such, the mechanistic interpretation of this finding is that high intrinsic proliferation rates are

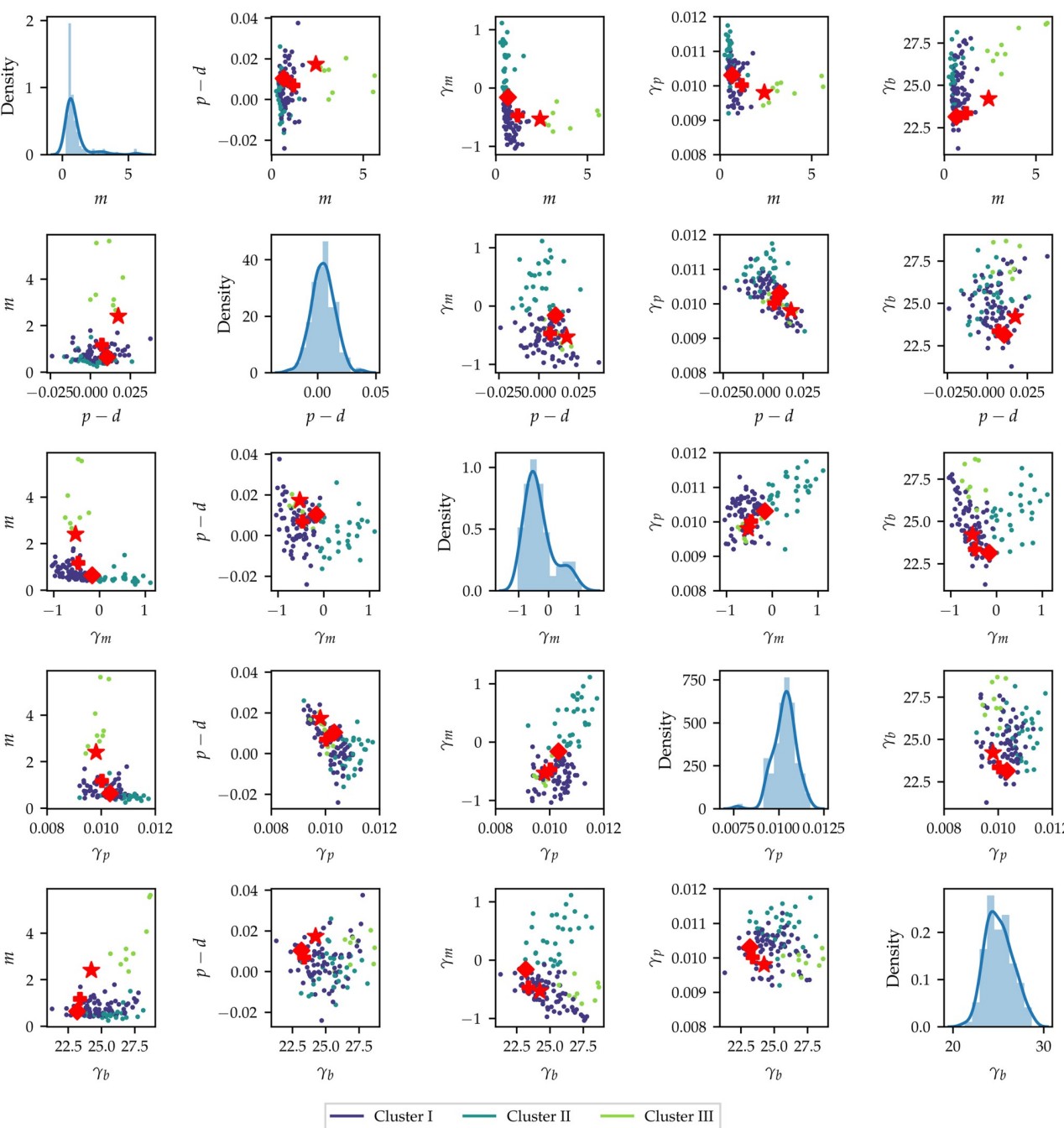

**Fig 6. Pairwise distributions of the posterior means for each of the knockdown experiments in the high-throughput screen.** Diagonal plots show the distribution of the posterior means of each of the parameters, while off-diagonal plots show the pairwise distributions of the different knockdown posterior means. The colour key denotes the different clusters found using $K$-means clustering with $K = 3$. Star denotes Mock, diamond CDC42 knockdown and plus CDH5 knockdown.

associated with lower density-dependent inhibition of proliferation. Fig 6 also suggests that there is a coupling between a higher increased intrinsic motility, $m$, and the bias parameter, $\gamma_b$: knockdowns in Cluster III also show the highest values of $\gamma_b$. This coupling is much less pronounced when the intrinsic net proliferation rate is compared to the density-dependent

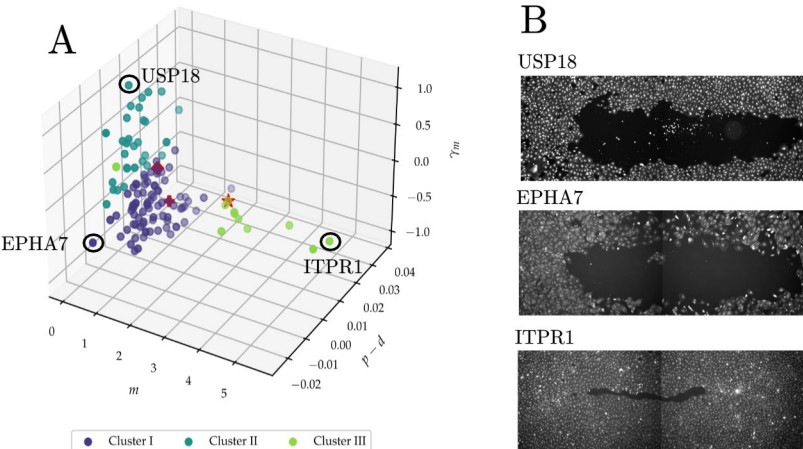

**Fig 7. Identifying functional subgroups among perturbations.** A: Three-dimensional scatter plot of the posterior means for the intrinsic motility parameter, $m$, intrinsic net proliferation parameter, $p - d$, and density-dependent motility parameter, $\gamma_m$, for each of the samples in the high-throughput screen. Star denotes Mock, diamond denotes CDC42 knockdown and plus denotes CDH5 knockdown. B, top to bottom: wound mask for one final condition of the USP18, EPHA7, and ITPR1 knockdowns, respectively.

crowding term, $\gamma_b$. In summary, analysis of the posterior means suggests that high intrinsic motility, but not high intrinsic net proliferation, are associated with a stronger sensing of local cell density gradients through the bias parameter, $\gamma_b$. Finally, the pairwise plots show that most of the functional differences between the knockdowns can be described in terms of the intrinsic motility rate, $m$, the net proliferation rate, $p - d$, and the density-dependent motility rate, $\gamma_m$. In the remainder of this section, we explore the impact of these parameters on the wound healing outcomes by analysing the observed cell count and wound area changes.

Fig 7 illustrates how the clusters can help tease apart the different mechanisms at play in wound healing for each of the gene knockouts. For each of the clusters, we select the knockouts with greatest dissimilarity with the other clusters. These are EPHA7, USP18 and ITPR1 for Clusters I, II, and III, respectively. While the EPHA7 and USP18 knockdowns have similar degrees of wound closure, the posterior means for the EPHA7 knockdown suggest that incomplete wound closure is mainly due to net cell death during the experiment, which is consistent with previous studies [36]. The high value for $\gamma_m$ for USP18 knockdown suggests that cells are unable to move into the wound area due to defects in cell adhesion. This indicates that USP18 might be required for cell adhesion turnover through deubiquitination. Moreover, ITPR1 is known to regulate apoptosis and our results confirm its dual role on cell motility and proliferation [37].

In summary, Bayesian inference with a large screen of gene knockdowns can be used to characterise the functional role of the genes under investigation given a sufficiently detailed mathematical model. Given such a characterisation (in our work, this is the division into functional clusters), experimentally testable hypotheses can be generated to further refine and explore our understanding of the genes under consideration.

**3.3.2 Density-dependent patterns of proliferation reproduce experimental patterns of cell count changes.** Changes in cell counts are a simple summary statistic to extract from scratch assay data [17, 20, 21] and provide an indication of the average net proliferation rate in a large population of cells, such as the cell monolayer used for scratch assays. However, computing an average net proliferation rate does not reflect more complex mechanisms, such as contact inhibition, that may influence the change in cell numbers throughout the experiment.

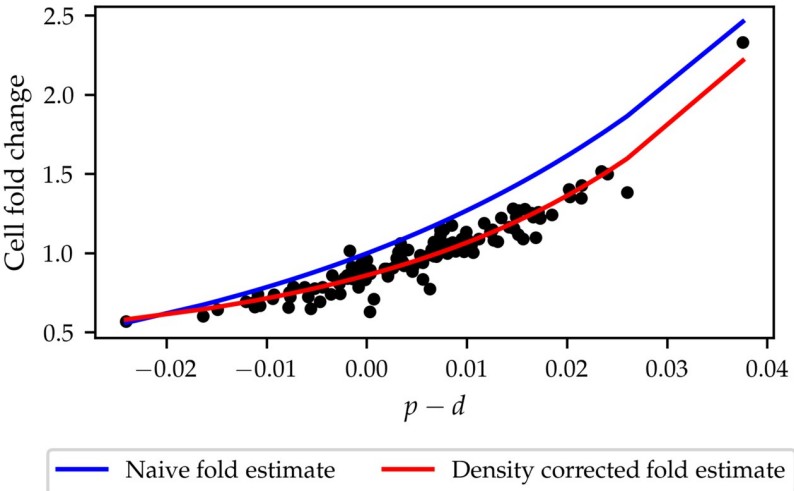

**Fig 8. Observed cell fold change as a function of intrinsic net proliferation rate, $p - d$.** When density-dependent effects are ignored (blue) line, the estimate of cell fold change as a function of net intrinsic proliferation overestimates cell fold change. When density-dependent effects are considered, model predictions accurately capture cell fold change.

In this section, we turn to the question of how density-dependent effects are observed in the high-throughput measurements in cell count changes for each of the knockdowns. Let $c_0$, $c_{24}$ denote the cell counts at $t = 0h$, $24h$ for a given knockdown. The fold change in cell count, $\mathcal{C}$, is given by $\mathcal{C} = c_{24}/c_0$. By the definition of the IBM, cell count changes occur as a function of the proliferation parameters $p$, $d$, and $\gamma_p$. In fact, Eq (4) relates the local proliferation rate to the intrinsic and density-dependent proliferation rates and hence relates the expected number of cells to the values of the different model parameters. In the case where density-dependent effects are ignored, *i.e.* $\gamma_p = 0$, one would expect that the expected value of the cell fold change, $\mathcal{C}$, is given by $\mathcal{C} = \exp(24(p - d))$, since $p - d$ is the net proliferation rate measured in $h^{-1}$. Fig 8 shows that this curve fails to describe the observed fold change in cell numbers across the screen, indicating that density-dependent effects affect fold changes in observed cell counts. In this work, density-dependent effects affect the proliferation rate through the sum in Eq (4). This sum depends on the cell locations at each simulation step, meaning that it cannot be determined from the available data. However, by assuming local cell density is constant across the assay, and by approximating local cell density as the average of the local cell densities at $t = 0h$ and $t = 24h$, this difficulty can be overcome. Letting $F$ be the area of the field of view, and $w_0$, $w_{24}$ the wound area at $t = 0h$, $t = 24h$, respectively, one can approximate the sum in Eq (4) by an integral such that

$$\gamma_p \sum_{i=1, i \neq n}^{N(t)} \exp\left( - \frac{\| \boldsymbol{x}_n - \boldsymbol{x}_i \|^2}{2\sigma^2} \right) \approx 2 r \gamma_p \sqrt{\pi} \left( \frac{c_0}{F - w_0} + \frac{c_{24}}{F - w_{24}} \right). \tag{16}$$

Under this approximation, a coarse-grained estimate for the fold change in cell numbers over the 24-hour experiment is given by

$$\mathcal{C} \approx \exp\left( 24 \left( p - d - 2 r \gamma_p \sqrt{\pi} \left( \frac{c_0}{F - w_0} + \frac{c_{24}}{F - w_{24}} \right) \right) \right). \tag{17}$$

For each sample in the high-throughput screen, we compute the value of $\mathcal{C}$ through Eq (17) and fit an exponential curve through the obtained fold estimates. In Fig 8 we show that the

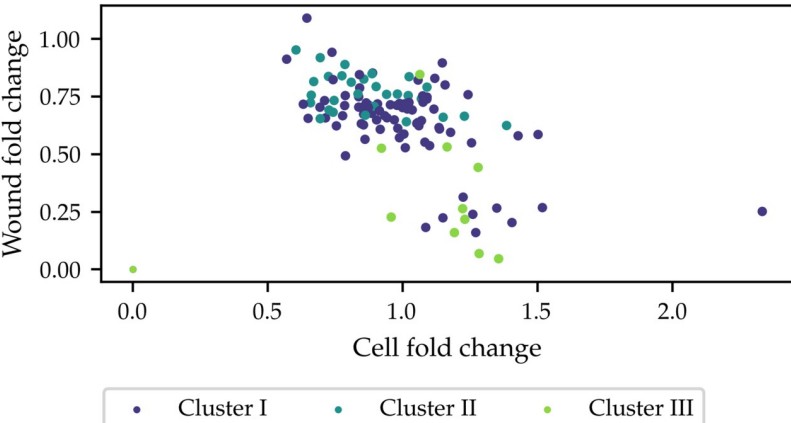

**Fig 9. Observed wound area fold change as a function of cell fold change.** Across the range of cell fold changes, starkly different rates of wound closure are observed.

density-corrected cell fold estimate predicts the observed cell fold change as a function of the intrinsic motility parameter well. We conclude that density-dependent effects contribute to the observed cell fold change by modulating with the intrinsic net proliferation rate and that the IBM can shed light on the nature of density-dependent effects for each of the knockdowns.

**3.3.3 Finding mechanistic contributions to wound closure.** While the fold change in cell counts is associated with the proliferation parameters, the fold change in wound size depends delicately and nonlinearly on all model parameters. We define the wound area fold change, $\mathcal{W}$, as $w_{24}/w_0$. Note that $\mathcal{W} = 1$ means that the wounds at $t = 0h$, $24h$ are equally large, and $\mathcal{W} = 0$ means the wound is fully closed. In this section we wish to understand the different roles of the various mechanistic terms in the model and explain the range of observed wound closure rates. In Fig 9, the observed wound area fold change is plotted against the observed cell fold change. While wound closure is not completely independent of increases in cell numbers, the data suggest that the dependence of wound closure on cell number increases is not as strong as might be expected if wound closure is thought to be primarily driven by proliferation: values of wound area fold change of approximately 0.75 are observed across knockdowns with 50% cell death all the way to 25% increases in cell counts, *i.e.* for net positive and net negative cell number changes, there can be similar amounts of wound closure. To investigate which mechanistic features of the model are key in wound closure, Fig 10 shows the observed wound area

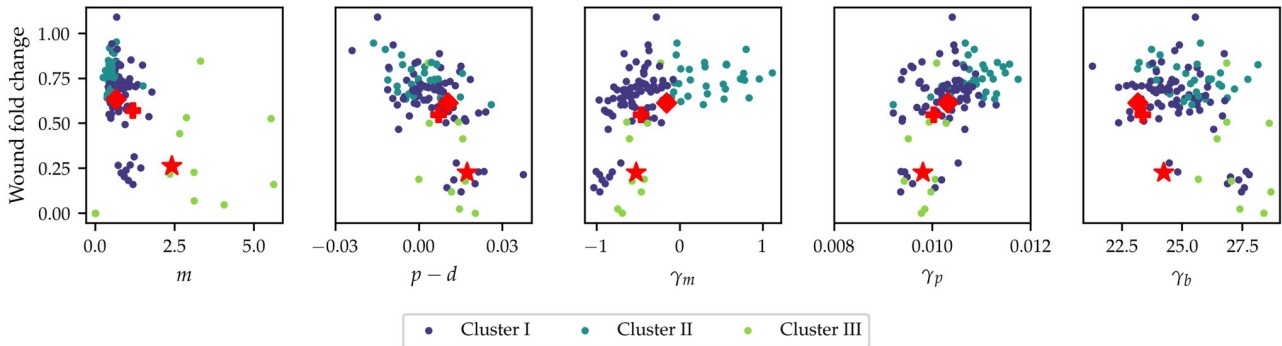

**Fig 10. Observed wound area fold change plotted against the different model parameters.** The colour key denotes the different clusters found using $K$-means clustering with $K = 3$. Star denotes Mock, diamond denotes CDC42 knockdown and plus denotes CDH5 knockdown.

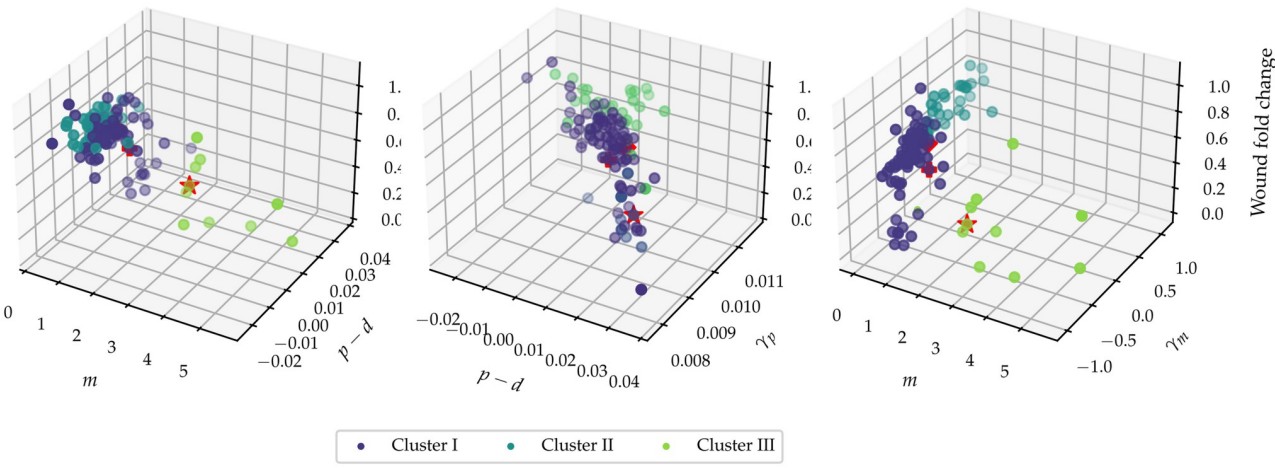

**Fig 11. Observed wound area fold change plotted as a function of combinations of posterior means.** Each three-dimensional plot shows the multivariate dependence of wound area fold change as a function of various parameters. The colour key denotes the different clusters found using $K$-means clustering with $K = 3$. Star denotes Mock, diamond denotes CDC42 knockdown and plus denotes CDH5 knockdown.

fold change, plotted against the different model parameters. Surprisingly, high intrinsic motility rates do not necessarily correlate with enhanced wound closure: the datapoints with labels corresponding to high intrinsic motility and density-enhanced motility all have much higher intrinsic motility posterior means than Mock, but their wound closure rates are not, in general, higher than Mock. At any given value of the intrinsic motility, $m$, a wide range of different wound closure extents are observed. In contrast, a stronger coupling is observed between the extent of wound closure and the density-dependent terms: when motility is strongly enhanced by local crowding, *i.e.* large, negative values of $\gamma_m$, very low values of wound area fold change are observed, whereas the wound area fold change is generally high when $\gamma_m$ is high. Likewise, when $\gamma_p$ is increased, wound area fold changes increase as well. These findings suggest that, rather than the intrinsic rates of movement and proliferation, it is the density-dependent interactions between cells that determine the degree to which wound healing is accomplished.

While Fig 10 provides information as to how the different model parameters influence wound closure, it is to be expected that there is an interplay between the intrinsic rates and their density-dependent counterparts. To investigate how different model parameters combine to give rise to wound closure, we investigate the dependence of wound closure on pairs of parameters. In Fig 11, wound area fold change is plotted as a function of the parameter pairs $(m, p - d)$, $(p - d, \gamma_p)$, $(m, \gamma_m)$.

Fig 11 suggests that high intrinsic motility rates can counteract low intrinsic proliferation rates to close the wound rapidly, but not vice versa. Moreover, the density-dependent mechanistic parameters have a profound impact upon wound area fold change. Between intrinsic motility, $m$, and density-dependent motility, $\gamma_m$, density-dependent interactions clearly play a crucial role. At the same time, across the range of values for the intrinsic net proliferation rate, $p - d$, the extent of density-dependent proliferation, $\gamma_p$, influences wound healing by increasing the extent of wound closure at similar levels of $p - d$.

## 4 Discussion and outlook

The aim of this work was to develop and showcase a method to address the task of performing Bayesian inference on a large dataset, where the likelihood is obtained through simulation of a detailed mathematical model and the nature of the data requires repeated simulation. In

essence, our method entails estimating the likelihood by repeatedly drawing subsamples from the data and estimating the likelihood function through this sampling scheme. The framework was developed in the context of a high-throughput scratch assay experiment, using a mathematical model with density-dependent interactions to describe the behaviours of the individual cells in the experiment, but the ethos of the approach is sufficiently general so as to apply to other large datasets and other computationally costly likelihood estimation methodologies.

The motivation for developing this framework stems from the fact that traditional high-throughput experiments often use simple summary statistics to understand and explain observed differences in the experiment. While these can lead to valuable information, a detailed mathematical model can provide further insights that are not captured by simple summary statistics. For example, this work focused on extracting detailed density-dependent interactions between cells from only considering spatial statistics in very sparse data. In this context, the identification of detailed density-dependent interactions allowed us to identify subgroups of genetic perturbations that have functionally different mechanistic behaviours. These differences help to explain and understand the different degrees of cell count changes and wound closure rates, which are summary statistics that are routinely collected from high-throughput experiments. Our findings identify different mechanistic behaviours that are consistent some with previous experimental measurements of wound healing after gene knockdowns. At the same time, the scale on which we are able to perform simulations, opens up the possibility of characterising in an automated way the functional role played by a large amount of gene knockdowns in the future. By using different, possibly more detailed mathematical models that are better tailored to a specific application, we conjecture that efficient Bayesian inference on a high-throughput experiment can be used to uncover previously undiscovered mechanisms—in the context of cell migration, or other applications.

We would like to highlight the importance of using a mathematical model that can faithfully identify and isolate mechanistic effects in the presence of observation noise or other factors. For instance, we noted in the process of carrying out the work that unless careful consideration was given to incorporating measured cell sizes in the assay, the model was unable to confidently identify cell-cell interactions, as the model components involved in estimating local cell density depend heavily on an accurate representation of cell size. Hence, by carefully extracting these detailed statistics, we can isolate the effect of cell-cell interactions and avoid the confounding factor that cell-cell interactions naturally occur over different spatial scales as a result of possibly different cell sizes. By probing how further mechanistic behaviours of the gene knockdowns affect these summary statistics, we conclude that density-dependent effects play a crucial role in wound healing.

There are a number of ways in which our method can be further improved going forward. Firstly, in this work we have used a simple method to choose the batch size, which amounts to analysing the systematic error and variability of the estimated parameter posterior distributions when different batch sizes are used, and choosing a value of the batch size where the variability plateaus while not incurring systematic errors in estimating model parameters. We remark that this amounts to a grid search through the batch size hyperparameter. Performing a grid search through algorithm hyperparameters prior to using the method on experimental data is a common approach in computational science to choose hyperparameters. In this case, the method yields additionally a rigorous uncertainty quantification of model predictions. The potential shortcoming of performing the hyperparameter grid search is that the computational cost of the grid search using different batch sizes might undo the computational gains from performing minibatch ABC-SMC. However, in many applications there may exist a need to perform ABC-SBC a great number of times. For example, in this work, there are 118 datasets, each of requires one ABC-SMC run. If, in each such ABC-SMC run, the computational time is

substantially reduced (and we note that the speed-up reported for *e.g.* the Mock dataset was a factor 11), the computational savings can still greatly exceed the extra cost of the hyperparameter grid search. At present, our method offers a computational improvement in such a setting. In future work, a different, perhaps more computationally efficient, method for choosing the minibatch size might be proposed, which would greatly extend and improve the applicability of our work.

More generally, the capacity of any ABC implementation to infer parameter values from data will depend on the specific computational model and the data available (for instance, through its spatial and temporal resolution, or its signal to noise ratio). For this work, data availability for several of the gene knockdowns was limited, and our work demonstrated that increasing the number of data points for these knockdowns would provide less variable estimates of model parameters, without incurring extra computational cost (by choosing a small batch size, the computational cost of the likelihood estimation does not increase, while the total variance does decrease). At the same time, a different choice of summary statistics might aid in the identification of some of the model parameters for which the posterior density is uninformative. The ability to do so will depend again on data availability, as well as a further understanding of the specific computational model (practical non-identifiability was also observed by Browning *et al.* [17] for the same parameters even though temporal resolution of the data was finer). These points nonwithstanding, uncertainty quantification demonstrated that even with modest amounts of data, minibatch ABC-SMC can reliably identify functional differences between different gene knockdowns. This opens the way to using ABC-SMC in understanding and interpreting the outcomes of high-throughput experiments as well as using ABC-SMC in such contexts to generate hypotheses regarding potential biological mechanisms that can be experimentally tested.

## Supporting information

**S1 Text. Supplementary information.**
(PDF)

## Author Contributions

**Conceptualization:** Simon Martina Perez, Heba Sailem, Ruth E. Baker.

**Data curation:** Simon Martina Perez.

**Methodology:** Simon Martina Perez.

**Software:** Simon Martina Perez.

**Supervision:** Heba Sailem, Ruth E. Baker.

**Visualization:** Simon Martina Perez.

**Writing – original draft:** Simon Martina Perez.

**Writing – review & editing:** Simon Martina Perez.

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
