## [Decision Letter · Decision Letter 0]

8 Mar 2022

Dear Mr Martina Perez,

Thank you very much for submitting your manuscript "Efficient Bayesian inference for mechanistic modelling with high-throughput data" for consideration at PLOS Computational Biology.

As with all papers reviewed by the journal, your manuscript was reviewed by members of the editorial board and by several independent reviewers. In light of the reviews (below this email), we would like to invite the resubmission of a significantly-revised version that takes into account the reviewers' comments.

We cannot make any decision about publication until we have seen the revised manuscript and your response to the reviewers' comments. Your revised manuscript is also likely to be sent to reviewers for further evaluation.

Sincerely,

Jennifer A. Flegg

Associate Editor

PLOS Computational Biology

Mark Alber

Deputy Editor

PLOS Computational Biology

Reviewer's Responses to Questions

**Comments to the Authors:**

Reviewer #1: The authors develop a new ABC method based on mini-batching (inspired by stochastic gradient descent in machine learning) in order to improve Bayesian inference for models with intractable likelihoods in the presence of a large number of independent datasets. Instead of trying to match on every dataset at each iteration, the comparison is based on a mini-batch of datasets randomly samples from all the datasets. The approach looks promising, but there are some issues/questions that need to be addressed before I can properly assess it:

1. Does the new approach outperform simply an analysis of a sensible subset of the dataset of the same size as the mini-batch? Eg some kind of stratified sampling to get a representative sample of initial conditions.

2. If the answer to 1 is yes, then the paper really needs to develop a principled method for selecting the mini-batch size for a given application. At the moment the paper tries a bunch of mini-batch sizes to see which is the best, but in practice if we try a bunch of different mini-batch sizes then we might lose any computational savings.

3. In the application the paper implicitly assumes that the same parameter value is responsible for generating all the datasets, but could there be any variation between parameter values for the different experiments? A related question is how well the model fits the data (including the datasets that were not included in the final mini-batch)? I don't think this was properly assessed in the paper.

4. This mini-batching is a bit different to whats in stochastic gradient descent, since there all the proposed updates are accepted. Whereas here mini-batches are rejected if they dont generate a small discrepancy. So is there some bias in the datasets that end up being selected in the mini-batch, ie some are accepted much more than others, since they are "easier" to match on, but may not be as informative about the parameters?

Reviewer #2: General impression:

The authors proposed an adaptation of the concept of mini-batching, which is widely used in the area of machine learning and which starts findings its way into other applications, to Approximate Bayesian Computation, with particular emphasis to systems biology. The method is evaluated on a stochastic model of single-cell motility and proliferation, which is used to model and understand a high-throughput scotch assay experiment.

I consider this a valuable contribution, as the availability of measurement data using high-throughput methods is indeed blessing and curse at a time. Learning from huge amounts of data is complicated and necessitates the development of new methods. ABC seems to be a methods which is particularly suited for combining it with mini-batching, due the stochastic nature of both concepts. The application example is interesting and important and showcases the potential of the method, although it is probably an edge case: It is just about to be large enough to see benefits from using a mini-batch approach, but not yet large enough to make mini-batch sampling the only way to cope with it (and the authors also don't claim that).

I feel like the manuscript is a bit in between a pure method contribution and an actual biological application. This has the advantage of showing a biologically relevant application example, but comes with the downside that e.g. the methodological part is not as thoroughly worked out as it could be the case.

I have a few (mostly technical) points, of which I hope they will help to improve the manuscript.

Major remarks:

I think the speed-up in lines L.376 - L.378, L. 411 is computed incorrectly. The authors run 4 generations of ABC-SMC, with mini batch size 10. That means only 40 datapoint (i.e., initial conditions) out of 118 are used at all. Typically, one uses in the mini batch methodology the notion of „epochs“, where one epoch is one pass through the dataset. In the present example, the model was not even „trained“ on the full dataset, so it was trained for less than one epoch. It is thus misleading to compare the computational speed-up with a full-batch approach using the whole dataset, since the model has not been informed by the full dataset. An approach using less than one epoch may work for this particular model and this particular dataset, but I doubt that this is a reliable approach in general. I would rather expect that for a general mini-batch-ABC-SMC application using N generations, the computational speed-up cannot exceed N, or, in this case, 4. It would be good if the authors used the notion of epochs and elaborated on this effect a bit.

The authors use ABC-SMC with a constant acceptance rate (fixed to 2,5%), which is not a common approach, I think. Usually, acceptance rates or effective sample sizes are adapted (e.g., decreased over time) in order to converge to the true posterior distribution. Different approaches how to choose the samples for the next generations exist (see e.g. [1], [2] or [3]). I would be interested if the approach used by the authors can converge to the true posterior at all (e.g., if there is a mathematical proof for this, ideally including the mini batch adaptation of the method). Alternatively, it might be worthwhile to test the minibatching approach on a (smaller) toy model and compare it to a well established method or (if possible) a ground-truth. In am particularly interested in this, as the number of generations used in this example seems comparably low to me.

Minor remarks:

At least I am not aware of minibatch approaches for ABC-SMC. However, minibatching has already been used for Bayesian methods such as MCMC (see [4]). Therefore, I think the work by Seita et al. is worth being mentioned by the authors, as it is on one hand close to their work, and on the other hand of interest for readers of this manuscript.

How are the mini batches drawn across different generations? Might it be possible to have one out of the 118 initial conditions used twice, despite using a mini batch size of 10 only?

On what infrastructure were the computational studies carried out? Was it a laptop or (as according to the acknowledgements) a cluster? How many nodes/cores were used and which types?

If possible, I would like to know how the computational speed-up reported by the authors compares to using established ABC-SMC toolboxes, which often allow for sophisticated parallelisation schemes, such as [5] or [6]. N.B.: Even if the speed-up by the method proposed by the authors does not exceed the one which can be achieved by using another toolbox, this does not reduce their contribution, as I consider the method being worth published also in that case, since it’s original and could additionally combined with other, already existing concepts for efficient computation.

L. 32 - L. 35: Is this really an exponential increase? It is not clear to me why this has to be the case… Would be happy if the author could add a reference for this. Alternatively, they could reword this part.

I think the font sizes in the figures labels and legends should be increased in general, as they are sometimes hard to read and there is usually enough space to do so.

In L.498, the authors speak of high-dimensional data. This feels a bit bizarre if the number of dimensions is five… The method used by the authors, which I think is appropriate, is also used on data with multiple hundred or thousand dimensions, where I consider this wording more appropriate. I would suggest to reformulate the sentence.

In L.510, the authors state that the „elbow method“ is commonly used in data science. In that case, I would like to see a reference.

Caption of Figure 10: The star, diamond, and plus are mentioned, but not shown in the figure.

The reference to Supplementary information S3 is incorrect, I think: This should be S2.3 instead.

Typo in the Abstract: „but this measurements“ -> „but these measurements“

References:

[1] Del Moral et al., An adaptive sequential Monte Carlo method for approximate Bayesian computation, 2012, Statistics and Computing 2

[2] Sik et al., Optimizing threshold-schedules for sequential approximate Bayesian computation: applications to molecular systems, 2013, Statistical Applications in Genetics and Molecular Biology

[3] Schälte et al., Efficient exact inference for dynamical systems with noisy measurements using sequential approximate Bayesian computation, 2020, Bioinformatics 36

[4] Seita et al., An Efficient Minibatch Acceptance Test for Metropolis-Hastings, 2018, Proceedings of the Twenty-Seventh International Joint Conference on Artificial Intelligence (IJCAI-18)

[5] Klinger et al., pyABC: distributed, likelihood-free inference, 2018, Bioinformatics 34

[6] Liepe et al., ABC-SysBio—approximate Bayesian computation in Python with GPU support, 2010, Bioinformatics 26

**Have the authors made all data and (if applicable) computational code underlying the findings in their manuscript fully available?**

Reviewer #1: Yes

Reviewer #2: Yes

PLOS authors have the option to publish the peer review history of their article (what does this mean?). If published, this will include your full peer review and any attached files.

Reviewer #1: No

Reviewer #2: No
---

## [Decision Letter · Decision Letter 1]

9 May 2022

Dear Mr Martina Perez,

We are pleased to inform you that your manuscript 'Efficient Bayesian inference for mechanistic modelling with high-throughput data' has been provisionally accepted for publication in PLOS Computational Biology.

Best regards,

Jennifer A. Flegg

Associate Editor

PLOS Computational Biology

Mark Alber

Deputy Editor

PLOS Computational Biology

Reviewer's Responses to Questions

**Comments to the Authors:**

Reviewer #1: Thanks to the authors for their efforts, my comments have been adequately addressed. I think there might be a minor typo "ABC-SBC" should be "ABC-SMC".

Reviewer #2: First of all, I want to thank the actors for their work and their detailed reply.

I think that only the points 1 and 2 from the previous list of questions and remark need a reply. Here they are:

1) Indeed, I have missed out the fact that during one generation, different mini-batches of data points will be used to propose different parameter vectors, i.e., I missed out a factor of N_gen in terms of "exploration" of the data set. Given the additional clarification, this eliminates my concerns and makes me agree with the author’s numbers for the computational speed-up.

2) I am well aware of the supplementary figure which showcases that the posterior didn’t change after only 2 or 3 generations. However, this does not mean that the method converged to the "true" posterior. It may just as well mean that the method got stuck and represents an incorrect distribution (which is, at least in my experience, a typical problem).  However, concerning fixed acceptance rates, the authors convinced me indeed that their approach is valid by their argumentation.

**Have the authors made all data and (if applicable) computational code underlying the findings in their manuscript fully available?**

Reviewer #1: None

Reviewer #2: Yes

PLOS authors have the option to publish the peer review history of their article (what does this mean?). If published, this will include your full peer review and any attached files.

Reviewer #1: No

Reviewer #2: No

---

## [Editor Report · Acceptance letter]

9 Jun 2022

PCOMPBIOL-D-22-00221R1 

Efficient Bayesian inference for mechanistic modelling with high-throughput data

Dear Dr Martina Perez,

I am pleased to inform you that your manuscript has been formally accepted for publication in PLOS Computational Biology. Your manuscript is now with our production department and you will be notified of the publication date in due course.

With kind regards,

Anita Estes
